# Adversarial Reprogramming Revisited

**Matthias Englert**[*]
University of Warwick
m.englert@warwick.ac.uk

**Ranko Lazić**[*]
University of Warwick
r.s.lazic@warwick.ac.uk

## Abstract

Adversarial reprogramming, introduced by Elsayed, Goodfellow, and Sohl-Dickstein, seeks to repurpose a neural network to perform a different task, by manipulating its input without modifying its weights. We prove that two-layer ReLU neural networks with random weights can be adversarially reprogrammed to achieve arbitrarily high accuracy on Bernoulli data models over hypercube vertices, provided the network width is no greater than its input dimension. We also substantially strengthen a recent result of Phuong and Lampert on directional convergence of gradient flow, and obtain as a corollary that training two-layer ReLU neural networks on orthogonally separable datasets can cause their adversarial reprogramming to fail. We support these theoretical results by experiments that demonstrate that, as long as batch normalisation layers are suitably initialised, even untrained networks with random weights are susceptible to adversarial reprogramming. This is in contrast to observations in several recent works that suggested that adversarial reprogramming is not possible for untrained networks to any degree of reliability.

## 1 Introduction

Elsayed, Goodfellow, and Sohl-Dickstein [2019] proposed *adversarial reprogramming*: given a neural network $\mathcal{N}$ which performs a task $F : X \to Y$ and given an adversarial task $G : \widetilde{X} \to \widetilde{Y}$, repurpose $\mathcal{N}$ to perform $G$ by finding mappings $h_{\text{in}} : \widetilde{X} \to X$ and $h_{\text{out}} : Y \to \widetilde{Y}$ such that $G \approx h_{\text{out}} \circ F \circ h_{\text{in}}$. They focused on a setting where $F$ and $G$ are image classification tasks, $X$ consists of large images, $\widetilde{X}$ consists of small images, and $h_{\text{in}}$ and $h_{\text{out}}$ are simple and computationally inexpensive: $h_{\text{in}}(\widetilde{x}) = p + \widetilde{x}$ draws the input $\widetilde{x}$ at the centre of the *adversarial program $p$*, and $h_{\text{out}}$ is a hard coded mapping from the class labels $Y$ to the class labels $\widetilde{Y}$. Then the challenge of adversarial reprogramming is to find $p$ such that $h_{\text{out}} \circ F \circ h_{\text{in}}$ approximates well the adversarial task $G$.

Remarkably, Elsayed et al. [2019] showed that a range of realistic neural networks can be adversarially reprogrammed to perform successfully several tasks. They considered six architectures trained on ImageNet [Russakovsky, Deng, Su, Krause, Satheesh, Ma, Huang, Karpathy, Khosla, Bernstein, Berg, and Fei-Fei, 2015] and found adversarial programs that achieve very good accuracies on a counting task, MNIST [LeCun, Bottou, Bengio, and Haffner, 1998], and CIFAR-10 [Krizhevsky, 2009]. In addition to the basic setting, they investigated limiting the visibility of the adversarial program by restricting its size or scale, or even concealing both the input and the adversarial program by hiding them within a normal image from ImageNet, and obtained good results.

Adversarial reprogramming can be seen as taking the crafting of adversarial examples [Biggio, Corona, Maiorca, Nelson, Šrndić, Laskov, Giacinto, and Roli, 2013, Szegedy, Zaremba, Sutskever, Bruna, Erhan, Goodfellow, and Fergus, 2014] to a next level. Like the universal adversarial perturbation of Moosavi-Dezfooli, Fawzi, Fawzi, and Frossard [2017], a single adversarial program is combined with every input from the adversarial task, but in contrast to the former where the goal is to cause

---

[*]Equal contribution.

36th Conference on Neural Information Processing Systems (NeurIPS 2022).

natural images to be misclassified with high probability, adversarial reprogramming seeks a high accuracy of classification for the given adversarial task.

Adversarial reprogramming is also related to transfer learning [Raina, Battle, Lee, Packer, and Ng, 2007, Mesnil, Dauphin, Glorot, Rifai, Bengio, Goodfellow, Lavoie, Muller, Desjardins, Warde-Farley, Vincent, Courville, and Bergstra, 2011], however with important differences. Whereas transfer learning methods use the knowledge obtained from one task as a base for learning to perform another, and allow model parameters to be changed for the new task, in adversarial reprogramming the new task may bear no similarity with the old, and the model cannot be altered but has to be manipulated through its input.

The latter features are suggestive of potential nefarious uses of adversarial reprogramming, and Elsayed et al. [2019] list several, such as repurposing computational resources to perform a task which violates the ethics code of system providers. However, its further explorations have demonstrated utility for virtuous deployments to medical image classification [Tsai, Chen, and Ho, 2020], natural language processing [Hambardzumyan, Khachatrian, and May, 2021, Neekhara, Hussain, Dubnov, and Koushanfar, 2019, Neekhara, Hussain, Du, Dubnov, Koushanfar, and McAuley, 2022], molecule toxicity prediction [Vinod, Chen, and Das, 2020], and time series classification [Yang, Tsai, and Chen, 2021]. In the last three works, adversarial reprogramming was achieved between different domains, e.g. repurposing neural networks trained on ImageNet to perform classification of DNA sequences, and of natural language sentiments and topics.

In spite of the variety of fruitful applications, it is still largely a mystery when adversarial reprogramming is possible and why. In the only work in this direction, Zheng, Feng, Xia, Jiang, Demontis, Pintor, Biggio, and Roli [2021] proposed the alignment $\|g\|_1 / (1/n \sum_i \|g_i\|_1)$ of the gradients $g_i$ of the inputs from the adversarial task with their average $g$ as the main indication of whether adversarial reprogramming will succeed. However, their experiments did not show a significant correlation between accuracy after reprogramming and the gradient alignment before reprogramming. The correlation was statistically significant with the gradient alignment after reprogramming, but that is unsurprising and it is unclear how to use it for predicting success.

Specifically, a central question on adversarial reprogramming is:

> Can neural networks with random weights be adversarially reprogrammed, and more generally, how does training impact adversarial reprogrammability?

First, addressing this question is important for assessing scope of two claims made in the literature:

***"[Adversarial] reprogramming usually fails when applied to untrained networks"* [Zheng et al., 2021].** In addition to networks trained on ImageNet, Elsayed et al. [2019] experimented using the same architectures untrained, i.e. with random weights, and obtained generally poor results. Similarly, the experimental results of Neekhara et al. [2019] and Zheng et al. [2021] for random networks are significantly worse than their experimental results for trained networks. However, they remarked that this was surprising given the known richness of random networks (see e.g. He, Wang, and Hopcroft [2016c], Lee, Bahri, Novak, Schoenholz, Pennington, and Sohl-Dickstein [2018]), and that it was possibly due to simple reasons such as poor scaling of the random weights.

***"The original task the neural networks perform is important for adversarial reprogramming"* [El-sayed et al., 2019].** Nevertheless a number experimental results including those of Tsai et al. [2020], Neekhara et al. [2022], Vinod et al. [2020], Yang et al. [2021], Zheng et al. [2021] demonstrated successful adversarial reprogramming between tasks that are seemingly unrelated (e.g. repurposing networks trained on ImageNet to act as HCL2000 [Zhang, Guo, Chen, and Li, 2009] classifiers) or from different domains, so the interaction between the original and adversarial tasks remains unclear.

Second, the question above is important because of implications for the following two applied considerations:

**Disentangling architecture and training as factors in adversarial reprogrammability.** Clarifying the respective bearings on adversarial reprogramming success of the network architecture and of the task (if any) it was trained for would improve decision making, either to maximise the success in beneficial scenarios or to minimise it in detrimental ones.

**Managing the cost of adversarial reprogramming.** Understanding when training the network is not essential, and when training for longer does not help or even hinders adversarial reprogrammability, should make it possible to control better the economic and environmental costs.

## 1.1 Our contributions

We initiate a theoretical study of adversarial reprogramming. In it we focus on two-layer neural networks with ReLU activation, and on adversarial tasks given by the Bernoulli distributions of Schmidt, Santurkar, Tsipras, Talwar, and Mądry [2018] over hypercube vertices. The latter are binary classification data models in which the two classes are represented by opposite hypercube vertices, and when sampling a data point for a given class, we flip each coordinate of the corresponding class vertex with a certain probability. This probability is a parameter that controls the difficulty of the classification task. These data models are inspired by the MNIST dataset because MNIST images are close to binary (many pixels are almost fully black or white). The remaining parameters are the radius of the hypercube, the input dimension of the neural network, and its width.

We prove that, in this setting, for networks with random weights, adversarial programs exist that achieve arbitrarily high accuracy on the Bernoulli adversarial tasks. This holds for a wide variety of parameter regimes, provided the network width is no greater than its input dimension. The adversarial programs we construct depend on the weights of the network and on the class vertices (i.e. the direction) of the Bernoulli data model, and their Euclidean length is likely to be close to the square root of the input dimension. We present these results in Section 2.

We also prove that, in the same setting, training the network on orthogonally separable datasets can cause adversarial reprogramming to fail. Phuong and Lampert [2021] recently showed that, under several assumptions, training a two-layer ReLU network on such datasets by gradient flow makes it converge to a linear combination of two maximum-margin neurons; and subsequently Wang and Pilanci [2022] obtained in a different manner the same conclusion under the same assumptions. We provide a simpler proof of a significantly stronger result: we show that the assumptions in Phuong and Lampert [2021] and Wang and Pilanci [2022] of small initialisation, and of positive and negative support examples spanning the whole space, are not needed; and we generalise to the exponential loss function as well as the logistic one. We then observe that, for any Bernoulli data model whose direction is in a half-space of the difference of the maximum-margin neurons, and for any adversarial program, the accuracy tends to $1/2$ (i.e. approaches guessing) under a mild assumption on the growth rate of the difficulty of the data model. We present these results in Section 3.

Both for the neworks with random weights and for the networks trained to infinity on orthogonally separable datasets, we then show that similar theoretical results can be obtained with a different kind of adversarial task, namely those given by the Gaussian distributions also of Schmidt et al. [2018]. The latter are mixtures of two spherical multivariate Gaussians, one per data class. Please see the appendix.

In the experimental part of our work, we demonstrate that, as long as batch normalisation layers are suitably initialised, even untrained networks with random weights are susceptible to adversarial reprogramming. Both the random weights and the batch normalisation layers are kept fixed throughout the finding of adversarial programs and their evaluation. Our experiments are conducted with six realistic network architectures and MNIST as the adversarial task. We investigate two different schemes to combine input images with adversarial programs: replacing the centre of the program by the image as was done by Elsayed et al. [2019], and scaling the image to the size of the program and then taking a convex combination of the two. Each of the two schemes has a ratio parameter, and we explore their different values. We find that the second scheme gives better results in our experiments, and that for some choices of the ratio parameter, the accuracies on the test set across all six architectures are not far below what Elsayed et al. [2019] reported for networks trained on ImageNet. Please see Section 4.

We conduct the same experiments also on the more challenging Fashion-MNIST [Xiao, Rasul, and Vollgraf, 2017] and Kuzushiji-MNIST [Clanuwat, Bober-Irizar, Kitamoto, Lamb, Yamamoto, and Ha, 2018] datasets, and obtain broadly similar results, however with lower test accuracies in several cases; they are reported in the appendix.

For a further discussion of relations with other works, please see the appendix.

We conclude the paper in Section 5, where we discuss limitations of our work and suggest directions for future work.

## 2   Random networks

**Basic notations.**   We write: $[n]$ for the set $\{1, \ldots, n\}$, $\|\boldsymbol{v}\|$ for the Euclidean length of a vector $\boldsymbol{v}$, $\angle(\boldsymbol{v}, \boldsymbol{v}')$ for the angle between $\boldsymbol{v}$ and $\boldsymbol{v}'$, and $\mathbb{H}^d$ for the $d$-dimensional unit hypercube $\{\pm 1/\sqrt{d}\}^d$.

**Two-layer ReLU networks.**   We consider two-layer neural networks $\mathcal{N}$ with the ReLU activation. We write $d$ for the input dimension, $k$ for the width, $\boldsymbol{w}_1, \ldots, \boldsymbol{w}_k \in \mathbb{R}^d$ for the weights of the first layer, and $a_1, \ldots, a_k \in \mathbb{R}$ for the weights of the second layer. For an input $\boldsymbol{x} \in \mathbb{R}^d$, the output is thus

$$\mathcal{N}(\boldsymbol{x}) \coloneqq \sum_{j=1}^{k} a_j \psi(\boldsymbol{w}_j^\top \boldsymbol{x}) \,,$$

where $\psi(u) = \max\{u, 0\}$ is the ReLU function.

**Random weights.**   In this section, we assume that the weights in $\mathcal{N}$ are random as follows:

- each $\boldsymbol{w}_j$ consists of $d$ independent centred Gaussians with variance $1/d$, and
- each $a_j$ is independently uniformly distributed in $\{\pm 1/\sqrt{k}\}$.

This distribution is as in Bubeck, Cherapanamjeri, Gidel, and Tachet des Combes [2021], and standard for theoretical investigations. It is similar to He's initialisation [He, Zhang, Ren, and Sun, 2015], with the second layer discretised for simplicity.

The variances of the weights are such that, for any input $\boldsymbol{x} \in \mathbb{R}^d$ of Euclidean length $\sqrt{d}$, each $\boldsymbol{w}_j^\top \boldsymbol{x}$ is a standard Gaussian, and for large widths $k$ the distribution of $\mathcal{N}(\boldsymbol{x})$ is close to centred Gaussian with variance $1/2$.

**Bernoulli data models.**   Adapting from Schmidt et al. [2018], given a hypercube vertex $\boldsymbol{\phi} \in \mathbb{H}^d$, a radius $\rho > 0$, and a class bias parameter $0 < \tau \le 1/2$, we define the $(\boldsymbol{\phi}, \rho, \tau)$-Bernoulli distribution over $(\boldsymbol{x}, y) \in \rho\mathbb{H}^d \times \{\pm 1\}$ as follows:

- first draw the label $y$ uniformly at random from $\{\pm 1\}$,
- then sample the data point $\boldsymbol{x}$ by taking $y\rho\boldsymbol{\phi}$ and flipping the sign of each coordinate independently with probability $1/2 - \tau$.

These binary classification data models on hypercube vertices are inspired by the MNIST dataset [Le-Cun et al., 1998]. The class bias parameter $\tau$ controls the difficulty of the classification task, which increases as $\tau$ tends to zero, i.e. as $1/\tau$ tends to infinity.

In this section and the next, we consider adversarial tasks that are $(\boldsymbol{\phi}, \rho, \tau)$-Bernoulli data models, and investigate variations of the parameters $\boldsymbol{\phi}$, $\rho$ and $\tau$.

**Adversarial program.**   In this section, we assume that $k \le d$, i.e. the network width is no greater than the input dimension, and we define an adversarial program $\boldsymbol{p}$ which depends on the weights of the network $\mathcal{N}$ and on the direction $\boldsymbol{\phi}$ of the Bernoulli data model.

With probability 1, for all $j \in [k]$, we have that $a_j \boldsymbol{w}_j^\top \boldsymbol{\phi} \ne 0$. Let us write $K^+$ for $\{j \in [k] \mid a_j \boldsymbol{w}_j^\top \boldsymbol{\phi} > 0\}$, and $K^-$ for $\{j \in [k] \mid a_j \boldsymbol{w}_j^\top \boldsymbol{\phi} < 0\}$. Then, for all $j \in [k]$, let:

$$\boldsymbol{p}'_j = \begin{cases} 0 & \text{if } j \in K^+, \\ -\sqrt{d/|K^-|} & \text{if } j \in K^-. \end{cases}$$

Since $k \le d$, with probability 1, the weights vectors $\boldsymbol{w}_j$ are linearly independent, i.e. the $k \times d$ matrix $\boldsymbol{W}$ whose rows are $\boldsymbol{w}_j$ has a positive smallest singular value $s_{\min}(\boldsymbol{W})$. Hence $\boldsymbol{p} \in \mathbb{R}^d$ exists such that

$$\boldsymbol{p}' = \boldsymbol{W}\boldsymbol{p} \quad \text{and} \quad \frac{\|\boldsymbol{p}'\|}{s_{\max}(\boldsymbol{W})} \le \|\boldsymbol{p}\| \le \frac{\|\boldsymbol{p}'\|}{s_{\min}(\boldsymbol{W})} \,. \tag{1}$$

The neurons in $K^+$ can be thought of as "helpful" for the adversarial task, and those in $K^-$ as "unhelpful". This definition of an adversarial program $\boldsymbol{p}$ ensures that its effect is to introduce as first-layer biases the entries of the vector $\boldsymbol{p}'$. They are 0 (i.e. do nothing) for every "helpful" neuron, and the negative value $-\sqrt{d/|K^-|}$ (i.e. reduce the contribution to the network output) for every "unhelpful" neuron.

From the inequalities in (1), we can expect $\|\boldsymbol{p}\| \approx \sqrt{d}$ if $k = o(d)$ (see the appendix for details).

**Expected reprogramming accuracy.** The accuracy for an adversarial task $\mathcal{D}$ is the probability that the sign of the output of the reprogrammed network conforms to the input label, i.e.

$$\mathbb{P}_{(\boldsymbol{x},y)\sim\mathcal{D}}\{y\mathcal{N}(\boldsymbol{p}+\boldsymbol{x}) > 0\} \, .$$

Our main result in this section is that, for sufficiently large input dimensions $d$, and under mild restrictions on the growth rates of the network width $k$, the radius $\rho$ and the difficulty $1/\tau$ of the Bernoulli data model, in expectation over the random network weights, the reprogramming accuracy is at least $(1 - C_1\gamma)(1 - \gamma^\dagger)$, which by tuning the probability parameters $\gamma$ and $\gamma^\dagger$ can be arbitrarily close to $100\%$. The growth rate restrictions indicate that networks with smaller widths can be reprogrammed for tasks with larger radii, but that networks with larger widths can be reprogrammed for tasks that are more difficult. The proof proceeds by analysing concentrations of the underlying distributions and involves establishing a theorem that bounds the reprogrammed network output; the details can be found in the appendix.

**Corollary 1.** *Suppose that*

$$k = \Theta(d^{\eta_{(k)}}) \, , \quad \rho = O(d^{\eta_{(\rho)}}) \, , \quad 1/\tau = O(d^{\eta_{(\tau)}}) \quad \text{and} \quad 1/\tau = \omega_d(1) \, ,$$

*where* $\eta_{(k)}, \eta_{(\rho)}, \eta_{(\tau)} \in [0, 1]$ *are arbitrary constants that satisfy*

$$\eta_{(\rho)} < 1 - \eta_{(k)}/2 \quad \text{and} \quad \eta_{(\tau)} < \eta_{(k)}/2 \, .$$

*Then, for sufficiently large input dimensions d, the expected accuracy of the adversarially reprogrammed network* $\mathcal{N}$ *on the* $(\boldsymbol{\phi}, \rho, \tau)$*-Bernoulli data model is arbitrarily close to* $100\%$.

## 3 Implicit bias

**Orthogonally separable dataset.** In this section, we consider training the network on a binary classification dataset $S = \{(\boldsymbol{x}_1, y_1), \ldots, (\boldsymbol{x}_n, y_n)\} \subseteq \mathbb{R}^d \times \{\pm 1\}$ which is orthogonally separable [Phuong and Lampert, 2021], i.e. for all $i, i' \in [n]$ we have:

$$\boldsymbol{x}_i^\top \boldsymbol{x}_{i'} > 0 \quad \text{if} \quad y_i = y_{i'} \, , \quad \text{and} \quad \boldsymbol{x}_i^\top \boldsymbol{x}_{i'} \leq 0 \quad \text{if} \quad y_i \neq y_{i'} \, .$$

In other words, every data point can act as a linear separator, although some data points from the opposite class may be exactly orthogonal to it.

**Gradient flow with exponential or logistic loss.** For two-layer ReLU networks with input dimension $d$ and width $k$ as before (but without the assumption $k \leq d$, which is not needed in this section), we denote the vector of all weights by

$$\boldsymbol{\theta} \coloneqq (\boldsymbol{w}_1, \ldots, \boldsymbol{w}_k, a_1, \ldots, a_k) \in \mathbb{R}^{k(d+1)} \, ,$$

and we write $\mathcal{N}_{\boldsymbol{\theta}}$ for the network whose weights are the coordinates of the vector $\boldsymbol{\theta}$.

The empirical loss of $\mathcal{N}_{\boldsymbol{\theta}}$ on $S$ is $\mathcal{L}(\boldsymbol{\theta}) \coloneqq \sum_{i=1}^n \ell(y_i \mathcal{N}_{\boldsymbol{\theta}}(\boldsymbol{x}_i))$, where $\ell$ is either the exponential $\ell_{\exp}(u) = e^{-u}$ or the logistic $\ell_{\log}(u) = \ln(1 + e^{-u})$ loss function.

A trajectory of gradient flow is a function $\boldsymbol{\theta}(t) : [0, \infty) \to \mathbb{R}^{k(d+1)}$ that is an arc, i.e. it is absolutely continuous on every compact subinterval, and that satisfies the differential inclusion

$$\frac{\mathrm{d}\boldsymbol{\theta}}{\mathrm{d}t} \in -\partial\mathcal{L}(\boldsymbol{\theta}(t)) \quad \text{for almost all} \quad t \in [0, \infty) \, ,$$

where $\partial\mathcal{L}$ denotes the Clarke subdifferential [Clarke, 1975] of the locally Lipschitz function $\mathcal{L}$.

Gradient flow is gradient descent with infinitesimal step size. We work with the Clarke subdifferential in order to handle the non-differentiability of the ReLU function at zero: $\partial\psi(0)$ is the whole interval $[0, 1]$. At points of continuous differentiability, the Clarke subdifferential amounts to the gradient, e.g. $\partial\mathcal{L}(\boldsymbol{\theta}) = \{\nabla\mathcal{L}(\boldsymbol{\theta})\}$. For some further background, see the appendix.

**Initialisation of network weights.**    In this section, we assume that the initialisation is

**balanced:** for all $j \in [k]$, at time $t = 0$ we have $|a_j| = \|\boldsymbol{w}_j\| > 0$; and

**live:** for both signs $s \in \{\pm 1\}$ there exist $i_s \in [n]$ and $j_s \in [k]$ such that $y_{i_s} = s$ and at time $t = 0$ we have $y_{i_s} a_{j_s} \psi(\boldsymbol{w}_{j_s}^\top \boldsymbol{x}_{i_s}) > 0$.

The balanced assumption has featured in previous work (see e.g. Phuong and Lampert [2021], Lyu, Li, Wang, and Arora [2021]). It ensures that it remains to hold throughout the training, and that the signs of the second-layer weights $a_j$ do not change (see the proof of Theorem 2). The live assumption (present probabilistically in Phuong and Lampert [2021]) is mild: it states that at least one positively initialised neuron is active for at least one positive input, and the same for negative ones.

**Convergence of gradient flow.**    Our main result in this section establishes that the early phase of training necessarily reaches a point where the empirical loss is less than $\ell(0)$, which implies that then every input is classified correctly by the network. Perhaps surprisingly, no small initialisation assumption is needed, however the proof makes extensive use of orthogonal separability of the dataset (see the appendix, which contains all proofs for this section).

**Theorem 2.** *There exists a time $t_0$ such that $\mathcal{L}(\boldsymbol{\theta}(t_0)) < \ell(0)$.*

This enables us to apply to the late phase recent results of Lyu and Li [2020], Ji and Telgarsky [2020], Lyu et al. [2021] and obtain the next corollary, which is significantly stronger than the main result of Phuong and Lampert [2021], extending it to exponential loss, and showing that assumptions of small initialisation, and of positive and negative support examples spanning the whole space, are not needed. The corollary establishes that each neuron converges to one of three types: a scaling of the maximum-margin vector for the positive data class, a scaling of the maximum-margin vector for the negative data class, or zero. The two maximum-margin vectors are defined as follows: for both signs $s \in \{\pm 1\}$, let $I_s := \{i \in [n] \mid y_i = s\}$, and let $\boldsymbol{v}_s$ be the unique minimiser of the quadratic problem

$$\text{minimise} \quad \frac{1}{2}\|\boldsymbol{v}\|^2 \quad \text{subject to} \quad \forall i \in I_s : \boldsymbol{v}^\top \boldsymbol{x}_i \geq 1 .$$

That a trajectory $\boldsymbol{\theta}(t)$ converges in direction to a vector $\widetilde{\boldsymbol{\theta}}$ means $\lim_{t \to \infty} \boldsymbol{\theta}(t)/\|\boldsymbol{\theta}(t)\| = \widetilde{\boldsymbol{\theta}}/\|\widetilde{\boldsymbol{\theta}}\|$.

**Corollary 3.** *As the time tends to infinity, we have that the empirical loss converges to zero, the Euclidean norm of the weights converges to infinity, and the weights converge in direction to some $\boldsymbol{\theta}$ such that for all $j \in [k]$ we have $|a_j| = \|\boldsymbol{w}_j\|$, and if $a_j \neq 0$ then*

$$\frac{\boldsymbol{w}_j}{\|\boldsymbol{w}_j\|} = \frac{\boldsymbol{v}_{\text{sgn}(a_j)}}{\sum_{\text{sgn}(a_{j'})=\text{sgn}(a_j)} a_{j'}^2} .$$

Thanks to homogeneity, the sign of the network output does not depend on the norm of the weights, only on their direction. Examining networks whose weights are directional limits as in Corollary 3 is therefore informative of consequences for adversarial reprogramming of long training. The following result tells us that, for any Bernoulli data model whose direction is in a half-space of the difference of the maximum-margin vectors, and for any adversarial program, the accuracy tends to $1/2$ provided that the difficulty $1/\tau$ of the data model increases slower than the square root of the input dimension $d$. The latter assumption is considerably weaker than in the results of Schmidt et al. [2018] on the Bernoulli data model, where the bound is in terms of the fourth root. The statement also tells us that this failure cannot be fixed by choosing in advance a different mapping from the class labels of the original task to the class labels of the adversarial task. Since we consider binary classification tasks here, the mapping can be represented by a multiplier $m \in \{\pm 1\}$.

**Proposition 4.** *Suppose network weights $\boldsymbol{\theta}$ are is in Corollary 3, class label mapping $m \in \{\pm 1\}$ is arbitrary, data model $\mathcal{D}$ is any $(\boldsymbol{\phi}, \rho, \tau)$-Bernoulli distribution such that $m \cos \angle(\boldsymbol{v}_1 - \boldsymbol{v}_{-1}, \boldsymbol{\phi}) < 0$, and adversarial program $\boldsymbol{p}$ is arbitrary. Then we have that*

$$\mathbb{P}_{(\boldsymbol{x}, y) \sim \mathcal{D}}\{m \, y \, \mathcal{N}_{\boldsymbol{\theta}}(\boldsymbol{p} + \boldsymbol{x}) > 0\} \leq \frac{1}{2} + \frac{1}{2} e^{-2d\tau^2 \cos^2 \angle(\boldsymbol{v}_1 - \boldsymbol{v}_{-1}, \boldsymbol{\phi})} .$$

# 4  Experiments

**Network architectures and initialisation.**  Our experiments[2] are conducted using the following six network architectures: ResNet-50 [He, Zhang, Ren, and Sun, 2016a], ResNet-50V2, ResNet-101V2, ResNet-152V2 [He, Zhang, Ren, and Sun, 2016b], Inception-v3 [Szegedy, Vanhoucke, Ioffe, Shlens, and Wojna, 2016], and EfficientNet-B0 [Tan and Le, 2019].

We use the networks exactly as implemented in Keras in TensorFlow 2.8.1 including the method for randomly initialising the trainable weights. For biases, this means they are initialised with $0$. For all other trainable weights, mostly, the Glorot uniform initialiser [Glorot and Bengio, 2010] is used in this implementation. EfficientNet is an exception, where many layers instead use a truncated normal distribution that has mean $0$ and standard deviation $\sqrt{2/\text{number of output units}}$.

All the networks we experiment with (ResNet-50, ResNet-50V2, ResNet-101V2, ResNet-152V2, Inception-v3, and EfficientNet-B0) involve batch normalisation layers [Ioffe and Szegedy, 2015]. Every such layer maintains a moving mean and a moving variance based on batches it has seen during training. The inputs are then normalised accordingly: they are shifted by the recorded mean and scaled by the inverse of the recorded variance. Note that the moving mean and variance values are not trainable, i.e., they are not subject to updates by the optimiser during training. During inference, the moving mean and variance values are no longer updated, and the normalisation is performed based on the last values recorded during training.

Crucially, in the default implementation of these networks, these moving mean and moving variance values are initialised as $0$ and $1$, respectively. Therefore, an untrained network initialised in this way will behave as if the batch normalisation layers were not present.

To obtain more sensible random networks, i.e., ones that can still make use of batch normalisation, we initialise the moving mean and variance of batch normalisation layers differently. We generate a batch of 50 random images (each pixel value is chosen independently and uniformly at random in the allowed range). This single batch is then fed through the random network and each batch normalisation layer records the mean and variance values it sees at its input (and normalises its output accordingly). The trainable weights of the network are not changed during this process.

We should note that, in addition to the moving mean and variance, batch normalisation layers can have *trainable* weights, by means of which the output mean and variance can be tuned. Specifically, such a layer may have trainable weights $\gamma$ and $\beta$, and will scale its otherwise normalised output by $\gamma$ and shift it by $\beta$. If present, these trainable weights are initialised as $1$ and $0$ respectively, and hence have no effect. Our initialisation procedure does not modify these trainable weights and they are therefore not used in our random networks.

For each network, after randomly initialising its weights as set out in Section 4 and its batch normalisation layers as described above, we keep it completely fixed: neither its weights nor its batch normalisation layers (i.e., their moving means and variances, and their weights if any) change in any way.

**Combining input images with adversarial programs.**  Our adversarial programs are colour images whose sizes match the expected input size of the networks. This is $224 \times 224$ for all networks except Inception-v3, where it is $299 \times 299$.

We use two different schemes to combine input images with the adversarial program. The first scheme, used by Elsayed et al. [2019], is to take the adversarial program and overwrite a portion of it by the input image. We do this in such a way that the input image is, up to rounding, centred in the adversarial program. We can vary the construction by scaling the input image up or down before applying this procedure. In particular, we use a parameter $r \in [0, 1]$ and scale the input image, using bilinear interpolation, in such a way that $r$ times the width of the adversarial program is equal to the width of the scaled image. That we focus on the width is not important because all our inputs and programs are square. An illustration is shown in Figure 1.

Our second scheme involves scaling the image to the same size as the adversarial program and then taking a convex combination of the two. We use a parameter $v \in [0, 1]$ to specify how much weight

---

[2]We are making code to run the experiments available at `https://github.com/englert-m/adversarial_reprogramming`.

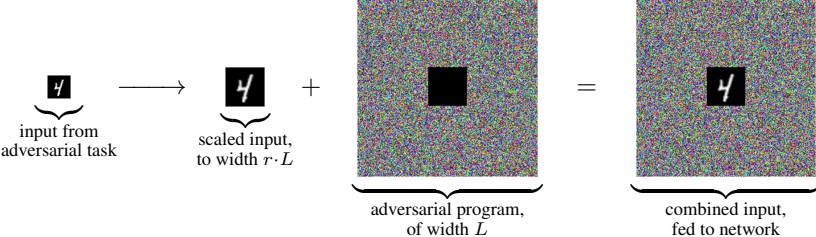

Figure 1: Scheme 1 for combining input images with adversarial programs. In this example, the width and height of the adversarial program are 224 and the parameter $r$ equals $2^{-20/9} \approx 0.214$, so the input image is scaled to width and height $r \cdot 224$ rounded, which is 48.

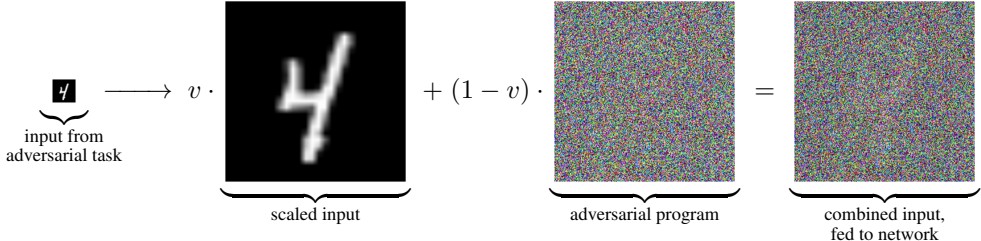

Figure 2: Scheme 2 for combining input images with adversarial programs. In this example, the sizes of the scaled input image and the adversarial program are $224 \times 224$. The parameter $v$ equals $2^{-40/9} \approx 0.046$, so the weight of the input image in the convex combination with the adversarial program is approximately $4.6\%$, which makes it faintly visible.

the input image should get in this convex combination. Specifically, the combined image is obtained by calculating $v \cdot I + (1 - v) \cdot P$, where $I$ is the input image and $P$ is the adversarial program. An illustration is shown in Figure 2.

**Adversarial task dataset.** Elsayed et al. [2019] evaluate adversarial reprogramming on random networks using the MNIST [LeCun et al., 1998] dataset. In other words, they were asking whether it is possible to repurpose a random network for the task of classifying the handwritten digits from the MNIST dataset. We use the same dataset, which consists of 60,000 training images and 10,000 test images, for our experiments. It is available under the Creative Commons Attribution Share-Alike 3.0 licence.

The networks we use classify inputs into 1,000 classes. We map the 10 labels of the MNIST dataset onto the first 10 of these classes.

For additional experimental results on the Fashion-MNIST and Kuzushiji-MNIST datasets, please see the appendix.

**Finding and evaluating adversarial programs.** Internally, we represent adversarial programs using unconstrained weights. We then apply a softsign function to the weights to map them into the range $(-1, 1)$, and further shift and scale the program such that the pixel values lie in the same range that is used for the input images. The program is initialised in such a way that after the application of the softsign function, each value lies uniformly at random in $(-1, 1)$.

We use the 60,000 training images to run an Adam optimiser [Kingma and Ba, 2015] with learning rate 0.01 and a batch size of 50 to optimise the unconstrained weights of the adversarial program. We report the accuracy on the 10,000 test images after 20 epochs, please see Figure 3.

The experiments were mainly run on two internal clusters utilising a mix of NVIDIA GPUs such as GeForce RTX 3080 Ti, Quadro RTX 6000, GeForce RTX 3060, GeForce RTX 2080 Ti, and GeForce GTX 1080. Depending on the network, optimising a single adversarial program for 20 epochs takes between 30 minutes and 1.5 hours on a standard desktop computer with two NVIDIA GeForce RTX 3080 Ti GPUs.

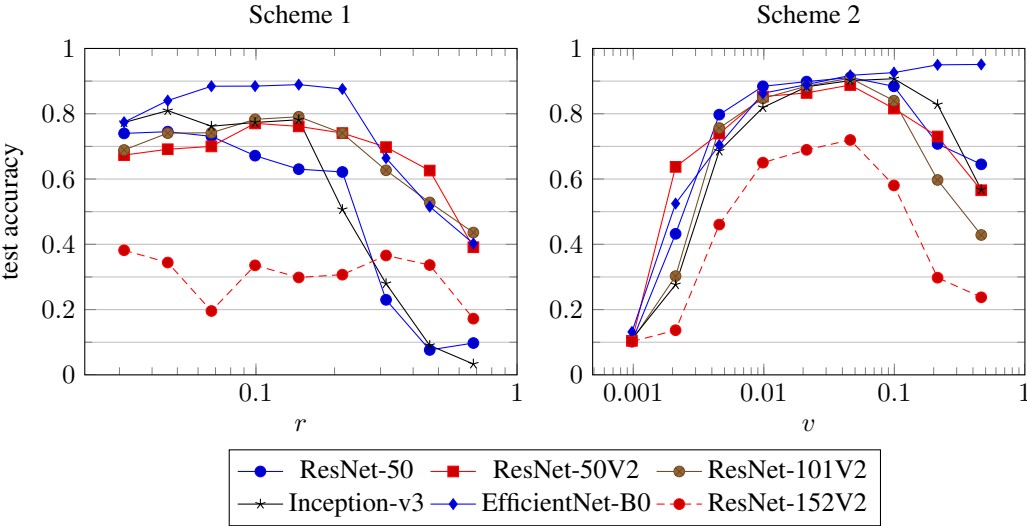

Figure 3: The accuracy achieved by the adversarial program on the MNIST test set for different parameters of the two schemes of combining input images with adversarial programs. The horizontal axes are logarithmic. The values plotted are averages over 5 trials, which are listed together with the standard deviations in the appendix.

We did not explore different optimisers and learning rates, since our first choices already resulted in suitable adversarial programs for these random networks. We only reduced the batch size to 50 after first trying 100, in order to reduce the requirement on GPU memory and be able to easily run the experiments on a wider range of hardware. However, we did extensively explore the two different schemes of combing input images with adversarial programs, and different values for the respective parameters $r$ and $v$. For each network, and each value of $r$ and $v$, we ran 5 experiments, each with a new random initialisation of the network, and are reporting the average of the test accuracy.

**Discussion.** Overall, the second scheme of combining input images with adversarial programs appears to give better and more reliable results in our experiments. For both schemes, the choice of parameters is important. Clearly, when $r$ or $v$ is 0, the input image is not visible to the network at all. On the other hand, when $r$ or $v$ is 1, there no longer is an adversarial program. In most cases, best results are achieved when the adversarial program is significantly larger (either by actual size in the first scheme, or in terms of pixel value ranges in the second scheme) than the input image.

In the second scheme in particular, we see accuracies on the test set which are lower, but not much lower than what Elsayed et al. [2019] reported for networks trained on ImageNet. For $v \approx 0.046$ for example, we see accuracies of 91.8%, 91.1%, 90.9%, 90.2%, 88.8%, 72.0% for EfficientNet-B0, ResNet-50, ResNet-101V2, Inception-v3, ResNet-50V2, ResNet-152V2, respectively. This suggests that, while training, say, on ImageNet may impact the possibility of finding suitable adversarial programs, such a training may be less important than previously thought and other factors are of significant importance.

# 5 Conclusion and future work

We proved the first theoretical results on adversarial reprogrammability of neural networks, in which we focused on architectures with two layers and ReLU activations, and on Bernoulli and Gaussian adversarial tasks. Provided the input dimension is sufficiently large, and for a wide variety of parameter regimes, our results show that: firstly, arbitrarily high reprogramming accuracies are achievable in expectation for networks with random weights; and secondly, reprogramming accuracies that are no better than guessing may be unavoidable for networks that were trained for many iterations with small learning rates on orthogonally separable datasets.

In the theoretical results that conclude arbitrarily high expected reprogramming accuracies, we assumed that the width of the random network is no grater than its input dimension, which is similar to the assumption on widths in e.g. Daniely and Shacham [2020] and enabled us to show existence of suitable adversarial programs by matrix inversion. Interesting directions for future work include relaxing this assumption, extending the whole theoretical analysis to deeper networks, and investigating derivation of adversarial programs by gradient methods.

It would also be interesting to consider more permissive data models; however, our theoretical results on the failure of adversarial reprogramming on networks that were trained to infinity rely on implicit bias properties of gradient methods, and in that area separability assumptions on training data are common and appear challenging to lift (see e.g. Lyu et al. [2021]).

The outcomes of our experiments, which are on six realistic convolutional network architectures designed for image classification, and on three adversarial tasks provided by the MNIST, Fashion-MNIST and Kuzushiji-MNIST datasets, are consistent with our theoretical results on high reprogramming accuracies. Both the experimental plots of test accuracies and the theoretical bounds of network outputs indicate existence of "sweet spot" maximising parameter values.[3] Since in the experiments, the network architectures have widths, depths and other features that are currently beyond our theoretical assumptions, and the adversarial programs are derived by gradient descent, their outcomes provide further motivation for extending the theory as suggested above.

A conclusion that emerges across our experimental results is that the EfficientNet-B0 architecture tends to be more susceptible to adversarial reprogramming, and the ResNet-152V2 architecture tends to be less susceptible, than the remaining four. We suggest for future work investigating the causes of this, as well as seeking to reprogram random networks for adversarial tasks that are more difficult than MNIST, Fashion-MNIST and Kuzushiji-MNIST. In the context of the latter, it would be interesting to consider also vision transformer [Dosovitskiy, Beyer, Kolesnikov, Weissenborn, Zhai, Unterthiner, Dehghani, Minderer, Heigold, Gelly, Uszkoreit, and Houlsby, 2021] architectures.

Another direction for experimental work is to verify that long training can cause adversarial reprogramming to fail.

## Acknowledgments and Disclosure of Funding

We are grateful to the anonymous reviewers, whose comments helped us improve the paper. We also thank Maria Ovens for linguistic advice.

This research was partially supported by the Centre for Discrete Mathematics and Its Applications (DIMAP) at the University of Warwick.

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
