# OpenReview forum: "Adversarial Reprogramming Revisited"
_NeurIPS.cc/2022/Conference — NeurIPS 2022 Accept_

### Official Review · Reviewer_Fq7W · 2022-06-29

**Rating:** 7
**Confidence:** 4
**Soundness:** 3 good
**Presentation:** 3 good
**Contribution:** 3 good

**Summary:**

The paper studies adversarial reprogramming (Elsayed et al 2019) that is the adversarial attack setting that can repurpose a network to conduct a task of interest. The paper provide theoretical proofs using simple random network and conclude that random networks can be adversarially repurposed as long as the input dimensionality is >= the network width. Elsayed et al 2019 managed to adversarially reprogram deep networks that are trained originally on ImageNet but failed to reprogram random deep networks. To address the possibility of training random networks in practice, the paper conducts experiments on multiple practical deep network architectures and show that the random networks can be repurposed to perform MNIST classification as long as batch normalization is suitably initialized, thus giving evidence that random networks could be trained in practice. The paper also explores two schemes for adding adversarial program (1) replacing the center by input images and (2) overlapping mixture of input images and adversarial program. The authors finds that the second scheme is more successful.

**Questions:**

Please see my above comments.

**Limitations:**

I think discussion of limitations is somewhat lacking. The authors needs to explain clearly the limitation of the work in a separate section or at the conclusion section. Please consult my comments on the strengths and weaknesses above.

**Strengths And Weaknesses:**

Strengths:
- Work provide theoretical proofs, though in reduced setting, of conditions where it is possible to adversarially reprogram neural networks.
- Authors explore two schemes of applying adversarial reprogramming of neural networks and demonstrate the superiority of one approach over the other.
- Authors provide empirical study on multiple network architectures


Weaknesses:
- Clarity of presentation: The theory in the paper could be somewhat overwhelming. I encourage the authors to give intuitions in addition to the theoretical proofs. To give an example, the authors prove that the two layer network can be reprogrammed given that the network width <= input dimensions. This conclusion may be confusing to some as a wider network has more capacity and may be regarded as more capable to do adversarial tasks. Maybe an intuition here is that the adversarial program dimensionally scales with input dimensions and in fact a larger input dimension allows more space to explore suitable adversarial programs that can repurpose the network.

- Theory limitations: To be able to provide theoretical proofs the authors needed to make simplifying assumption on the network architecture as well as the task. This although still an interesting contribution limit the impact of the work. I encourage authors can mention this limitation summary list of assumptions they needed to make. Perhaps also provide some suggestion for potential ways to extend the theory to more realistic networks and tasks.

- Discussion of batch norm: the discussion in the experimental section on batch normalization is not very clear. It is not 100% clear whether batch normalization is trained or left at initialization values. Also, changing the initialization scheme for batch normalization kind of violates the adversarial reprogramming setting where an adversary is given a network as is and one can only design the adversarial program.

- Experiments limitations: the authors use only one task (MNIST). I think authors needs to provide consistent evidence from at least two datasets to give confidence on the conclusions made. Further, experiments in Figure 2 uses one sample. The experiments need to be repeated from different seeds to get an estimate of variations (eg STD or SEM).

- It is unclear why scheme 2 of mixing adversarial programming with image is more successful. Is this because the dimensionality of useable adversarial program is higher than the scheme where the center of program is replaced with the image? It would be good to provide some explanation here.

---

> ### Author Response · Authors · 2022-08-02
> **Thank you very much for the detailed, constructive and inspiring review.**
>
> We hope that you will find useful our responses to the bullet points in the review under "Weaknesses", and then to the point in the "Limitations" section of the review:
>
> ---
>
> Regarding the assumption on the width in Section 2, we point out its source (namely our theoretical construction of adversarial programs by matrix inversion) in the new “Conclusion and future work” section (Section 5) in the revised version of the paper we have submitted, where we also remark that a similar assumption featured in a recent related work of Daniely & Shacham NeurIPS 2020.  Moreover, we identify seeking to relax it as a future direction, together with a different approach of obtaining adversarial programs (namely by gradient methods).
>
> To improve the clarity of presentation of the theory, we made several edits while revising the paper, including: spelling out the mathematical definition of reprogramming accuracy in Section 2; moving the main technical theorem in Section 2 (was Theorem 1) to the appendix, so that the focus of Section 2 is its corollary (now Corollary 1); simplifying the statement of Proposition 5 (now Proposition 4) which is the main result in Section 3 directly related to adversarial reprogramming, and adding further textual explanations before it; and producing new Figures 1 and 2 in Section 4 that hopefully clarify the two parameterised schemes of combining adversarial programs with arbitrary inputs.
>
> ---
>
> By moving our discussion of additional related work to the appendix, we gained space in the revised paper for the new “Conclusion and related work” section (Section 5), where we summarise the simplifying assumptions in our theory, compare them with our experimental setting, and suggest a number of directions for future work.  Encouraged by the consistency of our experimental results with our theoretical predictions even though the former break many of the restrictions in the latter, a large part of the future directions we propose are indeed towards extending the theory to more realistic networks and tasks.
>
> To demonstrate versatility of our theory, we extended the work to a second class of adversarial task, namely the Gaussian data models also of Schmidt et al., for which we obtained similar theoretical results, please see Appendix G of the revised paper.
>
> ---
>
> Thank you for this, we added to our description of how the batch normalisation layers are initialised (now in Appendix H.1 of the revised paper) the following paragraph: “For each network, after randomly initialising its weights as set out in Section 4 and its batch normalisation layers as described above, we keep it completely fixed: neither its weights nor its batch normalisation layers (i.e., their moving means and variances, and their weights if any) do not change in any way.”
>
> As we explain (now in the same appendix), leaving the batch normalisation means and variances at the default values of 0 and 1 amounts to removing these layers from the networks.  We would argue that initialising them randomly as we describe, together with initialising the rest of the network weights randomly, corresponds more closely to considering random instances of the architectures in question, which were designed to contain the batch normalisation layers.  In other words, we suggest that our simple random initialisation scheme for the batch normalisation layers (which does not depend in any way on the given adversarial task, or any other learnable task for that matter) allows us to investigate experimentially the “random richness” of these architectures, which would not be the case if we stuck exactly to the default initialisation as implemented in Keras.  If you think it worthwhile, we would be happy to add remarks along these lines to the paper.
>
> ---
>
> Thank you for this, we now report experimental results on two further datasets, Fashion-MNIST (images of clothing items) and Kuzushiji-MNIST (images of Japanese characters), please see Appendix H.2.  We now performed 5 experiments from different seeds per data point, and report average test accuracies as well as associated standard deviations (please see the new plots in Section 4 and Appendix H.2, and the new tables in Appendix H.3).
>
> ---
>
> Why Scheme 2 of mixing adversarial programs with images is more successful is a good question, however we are afraid we currently do not have a conclusive answer.  Seeking to explain that phenomenon merits future work!
>
> ---
>
> We added a new “Conclusion and future work” section largely for that purpose, in which we summarised the restrictions in our theoretical results, and the scope of our experiments.  The section also contains suggestions for future work, most of which are concerned either with bridging the gap between the current theory and realistic settings, or extending the experimental work to more challenging setups.

---

### Official Review · Reviewer_TjuY · 2022-07-11

**Rating:** 6
**Confidence:** 2
**Soundness:** 4 excellent
**Presentation:** 3 good
**Contribution:** 3 good

**Summary:**

This paper raises a fundamental question: what is the relationship between training and adversarial reprogrammability? The authors provide a theoretical analysis for adversarial reprogramming, where the conclusion shows that training has some but limited impact on reprogramming.

**Questions:**

Please see [Strengths And Weaknesses].

**Limitations:**

Please see [Strengths And Weaknesses].

**Strengths And Weaknesses:**

Strengths:

1. Reprogramming can have an important impact on many tasks, e.g. reprogramming for transfer learning, so a solid and fundamental investigation is promising.

2. The conducted experiments provide good support to the theoretical analysis.

Weakness:

1. The authors mention several application scenarios of reprogramming, but the connection between the theoretical analysis and the mentioned applications seems weak.

---

> ### Author Response · Authors · 2022-08-02
> **Many thanks for the encouraging review.**
>
> If we may, here is our response to the last point in the “Strengths And Weaknesses” part of the review.
>
> This is, to the best of our knowledge, the first work to provide a theoretical analysis of adversarial reprogramming.  As we point out in the introduction (Section 1), the latter can be seen as a powerful kind of adversarial attack, where one seeks perturbations (i.e., “adversarial programs”) that go beyond even universal targeted misclassification to induce specified target classes for a range of inputs (as given by the “adversarial task”).  Since theoretical explanations of adversarial examples continue to be a major challenge for the community (e.g. Bartlett et al. NeurIPS 2021 only recently proved that adversarial examples can be found by gradient steps in random constant-depth ReLU networks), it is commonplace in the literature to have simplifying restrictions such as the ones in our theoretical results: ReLU activations, one hidden layer, and a generic data model of adversarial tasks based on the “MNIST like” Bernoulli distribution on hypercube vertices of Schmidt et al. NeurIPS 2018.
>
> Nevertheless, we hope that you will find interesting that, in the submitted revised version of the paper, we extended our work to a second class of adversarial task, namely the Gaussian data models also of Schmidt et al., for which we obtained similar theoretical results (please see the new Appendix G).  There is also a new “Conclusion and future work” section (Section 5), in which we compare the theoretical assumptions with the experimental setup and identify directions for future work to bridge the gap.  We see it as encouraging that our experimental results are consistent with our theoretical predictions of successful adversarial reprogramming, moreover on architectures that are partly beyond the scope of the current theory.

---

### Official Review · Reviewer_iiJN · 2022-07-13

**Rating:** 6
**Confidence:** 3
**Soundness:** 2 fair
**Presentation:** 2 fair
**Contribution:** 2 fair

**Summary:**

This work studies the problem of "Adversarial Reprogramming" which is a task that is defined as follows. Suppose we have a machine learning model M that takes in inputs from a particular domain X to a label domain Y, while as a malicious actor imagine that we want to accomplish a task that takes in inputs from domain X' and outputs targets in domain Y'. Then the setting of adversarial reprogramming asks the question whether it is possible to find some simple functions h_in: X' -> X and h_out: Y -> Y' such that we can utilize h_out(M(h_in(x))) to solve this task. Note that finding a function h_out mapping from a discrete label space Y to Y' is a well studied problem in combinatorial optimization with several standard algorithms such as the Hungarian algorithm. Therefore, the main crux of the problem is to find the mapping h_in. Furthermore, in Adversarial Reprogramming a further assumption is made about h_in - that it be of the form of additive noise, i.e. h_in(x) = p + x, for some noise p which may depend on the model M. The goal then is to maximize accuracy on the task X' -> Y' without affecting the model parameters M (so one cannot fine-tune).

The authors study this problem for randomly initialized neural networks, i.e. when the model M is random. They show that in the 2-layer setting with the ReLU activation function and on a data distribution defined over the boolean hypercube one can find an adversarial program with arbitrarily high accuracy under several assumptions regarding network width etc. The authors also prove some theorems regarding when the dataset is orthogonally separable which I did not fully follow - e.g., Theorem 3 (are the convergence results about SGD or something else?). The authors also have some experiments which is hard to follow and I am not sure how their Theorems about 2 layer networks on boolean hypercubes translates to EfficientNets, Resnet-50 etc., on which they report experimental results.

**Questions:**

1) While the assumption about boolean hypercube may be a bit contrived, it is still a reasonable starting assumption in my opinion. However, it is not clear to me why are we interested in Adversarially reprogramming a random neural network? The motivation for that is not clear to me, an attacker would usually be interested in reprogramming an already pre-trained model.

2) The presentation of the paper is a little hard to follow, especially in Section 3  where someone not too familiar with the area will find it hard to follow. Moreover the consequence of Theorem 3 and Corollary 4 is not fully clear, what is the implication of this particular training dynamics?

3) The relation of the experimental section with the theoretical contribution of the work is unclear and again difficult to follow. Is it being validated that the theorem's proposed noise function is successfully finding some inputs on which a randomly initialized network has some high accuracy?

**Limitations:**

Yes

**Strengths And Weaknesses:**

Strengths: The paper seems to be about a topic that appears to be important, i.e. whether one can repurpose a model trained on a certain domain to another (potentially unethical or malicious) domain.

Weaknesses: The major weaknesses of the paper come from the several assumptions made about the Adversarial Reprogramming task:

Typically given a model M and a dataset domain (X, Y) a malicious actor would most likely wish to use M for a specific task (X', Y'): for example consider a scenario where an image classification model is trained to classify everyday objects (such as say Imagenet) but the malicious user wants to use it to predict whether people of a certain group are more criminally inclined than the others. However, the assumption made throughout is that "one can find an adversarial task" and the theorems also seem to work only for specific kinds of adversarial task (e.g., the theorem seems would not go through if the noise added in the boolean dataset task is some other quantity). However, this obtained adversarial task/dataset itself may not be as interesting or even malicious or meaningful to an attacker.

---

> ### Author Response · Authors · 2022-08-02
> **Thank you very much for the thoughtful review.**
>
> Here are our answers to the three questions:
>
> (1)  We believe that considering adversarial reprogramming of random neural networks is important for three reasons.  First, because it was investigated by several prior works who claimed that it cannot be done to any degree of reliability, which our results challenge and largely refute.  Second, in order to disentangle architecture and pre-training as factors in adversarial reprogrammability: random (i.e. untrained) neural networks can be regarded as “worst case”, so if a randomly initialised architecture can be successfully reprogrammed for an adversarial task, then that is an indication that it can be successfully reprogrammed regardless of whether and on what kind of dataset (related or unrelated to the adversarial task) it was pre-trained.  Third, to help manage the cost of adversarial reprogramming if it is desirable (such as in the various prior beneficial uses of it is a black box form of transfer learning that we cite).
>
> We have tried to make these points in the introduction (Section 1) of the paper.
>
> (2)  Indeed to fully digest Section 3, familiarity with some of the recent literature on implicit bias of gradient descent helps (e.g. Ji & Telgarsky NeurIPS 2020, Lyu & Li ICLR 2020, Lyu et al. NeurIPS 2021), in particular on training of ReLU neural networks by gradient flow.  As we point out, the latter is an idealisation of gradient descent with very small learning rates, which has been standard in this literature.  However, you are right that it therefore constitutes a limitation of our theoretical results in Section 3, and it is a question for future work whether e.g. stochastic gradient descent with realistic learning rates has the same properties.
>
> We hope that the main concluding part of Section 3 is more readable in the revised version of the paper which we have submitted, where we simplified the statement of Proposition 5 (which is now Proposition 4) and expanded the text before it.
>
> Theorem 3 and Corollary 4 (in the revised version, Theorem 2 and Corollary 3) may be important for readers who are familiar with Phuong & Lampert ICLR 2021 and Wang & Pilanci ICLR 2022 whose results we significantly improve there.  However, they are not directly about adversarial reprogramming, so may be regarded as auxiliary for obtaining Proposition 5 (now Proposition 4) by readers who are primarily interested in the latter.
>
> (3)  Your understanding is correct.  We apologise that the relation was not clear in our original submission, and we have tried to fix it in the revised version.  Section 4 now contains two new figures (Figures 1 and 2) that hopefully clarify the two schemes of combining the synthesised adversarial program with any given input from the adversarial task.  Furthermore, there is a new “Conclusion and future work” section (Section 5), in which we compare the theoretical assumptions with the experimental setup and identify directions for future work to bridge the gap.  We see it as encouraging that our experimental results are consistent with our theoretical predictions of successful adversarial reprogramming, moreover on architectures that are partly beyond the scope of the current theory.
>
> ---
>
> We also respond to a comment in the “Strengths And Weaknesses” part of the review, in case you will find it useful:
>
> You are right that the “attacker” chooses the adversarial task, and then needs to find an adversarial program (the “noise”) that when added to any input from the adversarial task makes the network output the classification that the attacker wants for that input (i.e. as specified in the adversarial task).
>
> Our theoretical results indeed apply only to a specific kind of adversarial task, namely those that can be described by a Bernoulli data model on hypercube vertices.  However, that is a generic class of “MNIST like” datasets introduced in the influential work of Schmidt et al. NeurIPS 2018, and our results allow arbitrary variations of the three parameters (direction, radius and class bias) as long as they satisfy the stated constraints.  So, by choosing those parameters, the attacker is able to “customise” the Bernoulli data model to make an adversarial task of interest to him or her.
>
> In the revised version, we added a comment in Section 2 that hopefully clarifies that the direction, radius and class bias parameters are variable, i.e. universally quantified in our results.  Moreover, we extended our work to a second class of adversarial task, namely the Gaussian data models also of Schmidt et al., for which we obtained similar theoretical results, please see Appendix G of the revised paper.
>
> In addition, we extended our experiments to two more datasets (i.e., adversarial tasks), namely Fashion-MNIST (images of clothing items) and Kuzushiji-MNIST (images of Japanese characters), and obtained also broadly positive results on reprogrammability of realistic random networks, please see Appendix H of the revised paper.

---

### Meta-Review · Area_Chair_RjCD · 2022-08-26

**Recommendation:** Accept
**Confidence:** Certain

**Metareview:**

This paper develops several theoretical results around when adversarial reprogramming will, and will not, be possible. They support these with experiments. As this is the first meaningful theoretical analysis of adversarial reprogramming, the paper is potentially impactful.

All reviewers support paper acceptance. Additionally, the paper appears to have been significantly improved over the course of the rebuttal period. I therefore recommend paper acceptance.

**Award:**

No

---

### Decision · Program_Chairs · 2022-09-14

Accept